# Childcare Balancing Policy in Japanese Corporations and Women's Fertility Intention

Yerong Zhao

Graduate School of Arts and Letters, Tohoku University, Kawauchi 27-1, Aoba Ward,
Sendai 980-8576, Miyagi, Japan; zhao.yerong.t2@dc.tohoku.ac.jp

**Abstract:** This study aimed to investigate the relationship between the childcare balancing policy and women's fertility intention in Japanese corporations. This paper constructed two logistic regression models based on data from the 2010 Japanese Life Course Survey of Youth to analyze the correlation between childcare balancing policies and women's fertility intentions. The binary logistic regression method was used. The results showed that women's fertility intention is negatively associated with the childcare balancing policy in Japanese corporations. This may be because the research object already had a child or children. The results indicate that the fertility intention of women who had a child or children was lower than those without children. This paper discovered that regular employees had higher fertility intentions than non-regular staff. This paper provides policymakers with valuable insights on establishing effective childcare policies to enhance women's fertility intentions.

**Keywords:** fertility intention; childcare balancing policy; Japanese corporations

## 1. Introduction

This paper focuses on the relationship between childcare balancing policies and women's fertility intentions. A childcare balancing policy in a company includes parental leave and childcare support policies, such as short work duration, flexible work hours, exemption from overtime, and reduced work hours for men (Suruga and Zhang 2003). The study's results are vital for policymakers and future women's fertility rates. It summarizes the declining birth rate in Japan and explains its reason from the perspectives of economic and social change and changes in women's values. Furthermore, it explores the employment conditions of women after marriage or childbirth in Japan. Thereafter, it introduces childcare policies in Japan, both by corporations and the government, such as childcare allowance and nursery school. Based on the above points, previous studies were analyzed, which led to the research question and hypotheses. This study addresses the analysis conducted through binary logistic regression and explains the results.

It must be noted that the childcare balancing policy differs from the parental leave policy, which will be discussed in the literature review.

### 1.1. The Declining Birth Rate in Japan and Its Reasons

Japan's total population has reached an all-time low in recent years.[1] This is because Japan faces the dual challenge of a declining birth rate and an aging population.

In developed countries, socio-economic changes and economic reconstruction give rise to a declining birth rate. Although it improves the life and economy of the people, under the diverse culture and welfare environment, it is difficult to build a family since, for instance, people tend to get married later or not at all.

Several factors affect the declining birth rate. In developed countries, economics could be a reason, such as the high cost of childbearing and childcare (Nargund 2009). The direct factors include late marriage, non-marriage, and late birth. Furthermore, women's educational achievements and employment rates, family income and poverty rates, and childcare facilities influence the birth rate (Sato 2008; Date and Shimizutani 2005).

In recent years, the process and progress of gender equality has been distinguished in Japan. Few women would quit their jobs because of marriage or maternity (Kuga 2018). Women need to achieve other goals besides being a wife and mother. Therefore, giving birth becomes a significant problem (Imada and Ikeda 2006).

Several other factors impact Japan's declining birth rate. Macdonald and Sasai (2008) offered a conception of "risk avoidance;" that is, being a part of the stable labor market could bring a high risk for young individuals; therefore, to avoid the risk, young individuals choose low-risk activities, such as investing in human capital, improving educational attainment, or gaining more work experience. Therefore, they invest sufficient energy through long working hours, which is adverse for building a family, leading to a low birth rate. In addition, if a family's economy is unstable, it is difficult to raise a child (Yamamoto and Kanda 2008).

After the second baby boom, Japan attained the lowest birth rate and total fertility rate in 2019.[2] According to Kawamoto (2018), the government published the "Next-generation upbringing support measures," such as supporting young generations' job-hunting and the use of parental leave. However, the birth rate continued to decline annually; thus, in 2007, a childcare balancing policy was issued.

### 1.2. Women's Employment

In Japan, women do most of the housework (Shinada 1996; Iwama 1997; Kan and Hertog 2017), especially bearing and raising a child. Therefore, the government focused on a work–life balance policy. A nationwide survey showed that dual-work families experience more work–family conflicts than other families (Pleck et al. 1980).

Women's employment in Japan has always been in the M shape. However, in recent years, the bottom of the M has risen more than before. This means that more women are increasingly choosing to work than leave their careers for marriage or having a child.

Therefore, for promoting women's employment and fertility intention and diminishing the work–life conflict, the Cabinet Office, in 2007, issued the childcare balancing policy in the workplace.

The childcare balancing policy encompasses several specific policies. For instance, Suruga and Zhang (2003) summarized that parental leave, childcare support policies, such as short work duration, flexible work hours, exemption from overtime, the abundance of childcare nursery schools, and childcare allowance, along with reduced work hours for men, are provided in most companies' policies. Kohara and Maity (2021) discussed that family-friendly policies include exemption from night shifts for people who attend to childcare or parental care and careful consideration when transferring such employees to a new place.

However, the childcare balancing policy is not statutory in companies. About 40.6% of companies in Japan have any one of the childcare balancing policies as per the 1999 Basic Survey on Women's Employment Management, and the rate of leading in short-time working is the highest at 29.9% (Imada and Ikeda 2004).

According to the 2016 Basic Survey on Equal Employment (Ministry of Health, Labor and Welfare 2016), the acquisition rate of parental leave[3] for women and men is 81.8% and 3.16%, respectively. However, there is no clear data about the acquisition rate of the childcare balancing policy.

### 1.3. The Problem in Previous Literature and the Research Question

Previous studies examined the relationship between women's employment and childcare balancing policies, such as how the policies positively influence women's work after marriage or childbirth, or how they impact the rate of women in authoritative positions. However, few studies have explored the relationship between childcare balancing policies and fertility intention, such as Murakami (2019) whose research on the relationship between the childcare balancing policy and women's fertility intention was not significant. However, the research method and object of the present study are different from Murakami (2019).

Originally, the childcare balancing policy was proposed to increase women's employment rates and children's birth rates. Nevertheless, most studies have focused on employment and ignored the birth rate. Therefore, this study examines the relationship between the childcare balancing policy and women's fertility intention. This research is significant and has social value. It is helpful for policymakers because they understand whether the policies affect fertility intention and the required amendments. Furthermore, by examining the relationship between the childcare balancing policy and women's fertility intention, the childcare balancing policy could be added as a controlled variable in future studies.

As Yamaguchi (2005) stated, the actual birth behavior is strongly related to fertility intentions. Fertility intentions play an important role in real birth behavior (Murakami 2019).

## 2. Literature Review

### 2.1. Women's Employment and Childcare Balancing Policy

Family–work conflicts are negatively associated with job and life satisfaction (Aryee et al. 1999). Due to extended working hours and content, women employees' productivity and hours, which they previously would have devoted to their families, have increased (Vasumathi 2018). Since women have the main responsibility for childcare in society, the dual roles make it difficult to balance work and family. To avoid such conflicts, some women choose not to give birth. Work–life balance policies were proposed to solve this problem.

After marriage or childbirth, due to the unequal division of labor at home, women bear the responsibility of childcare and increasingly devote time to the family. Some women pause or end their careers after childbirth. To avoid this, social support at work, in the form of the organization's family-friendly policies, is critical for women to continue working and contributing in the workplace (Marcinkus et al. 2007). Allen (2001) proposed that employees who found their company to be less family-supportive had less job satisfaction and experienced more work–family conflicts than those who found their company to be supportive.

Therefore, work–life balance or family-friendly policies include parental leave, flexible work hours, short work durations, and exemption from overtime (Wakisaka 2011).

#### 2.1.1. Parental Leave and Women's Employment

The parental leave policy was implemented in 1992; however, it could not penetrate most companies. Therefore, in 1999, the revised childcare leave policy was legally implemented. In 2010, 68.3% of companies had a parental/childcare leave policy (Wakisaka 2011). According to Zhang and Managi (2020), in 2017, the Japanese government increased the duration of paid parental leave, which the maximum of paid parental leave is 10 and 6 months and 6 months.

Companies with parental leave policies have more female employees since they can continue to work after childbirth (Tomita 1994). In companies where the parental leave policy can be utilized easily, the rate of women continuing to work after marriage is high. Furthermore, in areas with more nursery schools, the rate of women continuing to work after childbirth is high (Higuchi et al. 2016). After 2000, in companies with parental leave, the rate of quitting after childbirth has been statistically significantly low (Sato and Ma 2008). Companies are actively using female employees (Kawaguchi 2011).

This indicates that the parental leave policy is positively associated with women's retention rate.

#### 2.1.2. The Relationship between the Work–Life Balance Policy and Women's Employment

The essence of the work–life balance policy is that women can continue working, irrespective of the numerous issues faced at work and at home. Individuals who can achieve a work–family balance accomplish their value at the workplace and achieve their roles at home (Sato 2007). According to the Cabinet Office (2013), the acquisition rate of parental leave for females was 87.8% in 2011.

Abe (2007) proposed that the more a company implemented the work–life balance policy, the more productivity employees had. Tomita (1994) used the "Questionnaire survey on the actual conditions and issues of women's employment and labor" (in Osaka in 1993), and found that parental leave, nursery school in the company, short working hours, and half-day paid leave were positively associated with the rate of women continuing to work after childbirth.

According to Takeishi (2006), the application of work–life balance policies did not reduce the number of female employees. Numerous studies suggest that companies with family-friendly policies have a positive association with employees' work outcomes and attitudes and the female employment rate (Saito 2017; Kawaguchi and Kasai 2013; Suruga and Zhang 2003).

### 2.2. Factors Related to the Fertility Intention of Women

Childbearing motivations depend on personal characteristics and are shaped by childhood, adolescence, and early adult life experiences (Miller 1992).

By proposing a work–life balance, it is easy to decrease work–family conflicts, such as flexibility in family and work hours (Hill et al. 2001). Nevertheless, in the survey about work–life balance, when the work–life balance policy focuses on "work," "life" is somewhat neglected (Pichler 2008).

#### 2.2.1. Husband-Related Factors

The policies could release the burden of housework and childcare for women by making the men participate (Mizuochi 2011). Fujino (2006) proposed that when the wife works as a non-regular employee or zero-paid housewife, the husband's participation in housework and childcare would increase the number of children. Nishioka and Hoshi's (2009) research examined whether the work–life balance policy for men would affect women's fertility intention and found that men's participation in housework and childcare influenced women's fertility intention and the number of children. In addition, Meguro and Nishioka (2001) determined because the burden of housework and childcare worsened a female's fertility intention, the less the husband participated in housework and childcare, the lower the fertility desire.

#### 2.2.2. Age

Women's average childbearing age has increased over the past 20 years (Bray et al. 2006). The advanced age of women is negatively associated with birth outcomes (Roberts et al. 2011). Hence, with an increase in age, women's fertility intention declines (Tsuya 1999). Therefore, age has a strong connection with women's fertility intention (Westoff 1990).

#### 2.2.3. Financial Factors

There are various finance-related reasons for women's fertility intention, such as educational attainment (Axinn and Barber 2001; Basu 2002; Rindfuss et al. 1980; Rindfuss et al. 1996; Berrington and Pattaro 2014) and living with their parents (Tsuya 1999). The role of education is to impart the family's values and aspirations, similar to a workplace (Rindfuss et al. 1980). The high educational achievement of women is one of the most important factors affecting fertility intentions (Axinn and Barber 2001; Basu 2002; Rindfuss et al. 1980). Better-educated women may assume less traditional roles than less-educated women (Rindfuss et al. 1980). In addition, education has an indirect influence on birth (Rindfuss et al. 1980). Women with college degrees show a great shift toward childbearing because they have begun to pursue their careers (Rindfuss et al. 1996).

In Japan, the family's overall income has less impact on women's fertility intention than the husband's income concerning the first baby (Iwama 2004; Murakami 2019). Living with parents is positively associated with women's fertility intention. Even if women work outside, their parents could assist with childcare (Tsuya 1999).

### 2.2.4. Employment Status

For women who have two children, irrespective of regular and non-regular employment, the childbearing intention is higher than that of unemployed women (Matsuda 2019). Furthermore, a regular employee's fertility intention may be higher than a non-regular employee's intention because of a stable income (Murakami 2014). While childcare costs affect the birth rate for non-employed women, it does not impact their employed counterparts (Blau and Robins 1989). However, previous studies did not examine how the work–life balance policy influences regular and non-regular female employees' fertility intentions.

### 2.2.5. Hypotheses

There has been extensive research regarding the association between childcare balancing policies and women's employment. However, studies on the association between childcare balancing policies and women's fertility intentions are limited. Furthermore, studies on the relationship between employment status and women's fertility intentions have been extensively investigated. However, studies on how the childcare balancing policy influences regular and non-regular employees' fertility intentions are rare. Therefore, it is important to examine the relationship between childcare balancing policy and women's fertility intention. Excluding factors such as job stability and salary, understanding whether the policy affects regular and non-regular employees differently is helpful for policymakers and employers.

Although there are numerous factors affecting women's fertility intentions, the association between the childcare balancing policy and fertility intention is unexplored by previous studies. Furthermore, the policy was originally proposed because the government hoped that it would improve women's employment rate and relieve the declining birth rate. The birth rate is strongly related to women's fertility intentions (Yamaguchi 2005; Murakami 2019). Thus, hypothesis 1 is as follows:

**H1:** *There is a positive association between the childcare balancing policy and women's fertility intention.*

Previous studies have examined the relationship between employment status and women's fertility intention; however, the interaction term (childcare balancing policy *employment status) and whether the childcare balancing policy affects regular and non-regular female employees' fertility intention is unexplored. Thus, Hypothesis 2 is as follows:

**H2:** *Regular female employees' fertility intention is higher than non-regular employees under the childcare balancing policy.*

### 3. Data and Methods

*3.1. Data*

The Japanese Life Course Panel Survey of the Youth (JLPS-Y) data was used in this study. The JLPS was conducted annually from 2007–2012. The survey posed questions on various topics, such as individuals' occupation, family, education, political consciousness, and health. Participants were aged between 20–34 years, in 2006, when the survey was conducted.

The JLPS-Y from 2010 was chosen to examine Hypotheses 1 and 2. The survey was conducted in 2006 with a total sample size of 3367. There were 2974 variables in this data set in the whole panel survey. The survey method was mail delivery and door-to-door collection. The number of valid responses was 2121 (79%). The data set was chosen because it encompassed questions on the dependent and independent variables. Other data sets, covering related questions, were limited. For the analysis, samples were restricted to married women who were employed. After restricting the samples and excluding the missing values, the final sample size was 276.

Wave4, 2010 was adopted to analyze the data set.

*3.2. Variables*

Table 1 contains the descriptive statistics for the variables used in this analysis. For the dependent variable, the question posed regarding individuals' fertility intention was "Do you want a child?" with a binary response choice of 1 = yes and 2 = no. The response was coded as a dummy variable, with the values 0 = no and 1 = yes. For simplicity, this variable was named "wantbaby."

**Table 1.** Descriptive statistics of Analysis 1.

| | Percentage (%) | | |
|---|---|---|---|
| Fertility desire | | | |
| Yes | 63.77 | 176 | |
| No | 36.23 | 100 | |
| Employment status | | | |
| Regular employee | 47.10 | 130 | |
| Non-regular employee | 52.90 | 146 | |
| Workplace | | | |
| Not applicable | 14.13 | 39 | |
| Not very applicable | 16.30 | 45 | |
| Somewhat applicable | 36.59 | 101 | |
| Very applicable | 32.97 | 91 | |
| Income | | | |
| Approximately 200,000 | 0.36 | 1 | |
| Approximately 300,000 | 8.33 | 23 | |
| Approximately 400,000 | 14.13 | 39 | |
| Approximately 500,000 | 27.54 | 76 | |
| Approximately 700,000 | 31.16 | 86 | |
| Approximately 1,000,000 | 13.41 | 37 | |
| Approximately 1,500,000 | 3.62 | 10 | |
| Approximately 2,000,000 | 0.72 | 2 | |
| Over 2,250,000 | 0.72 | 2 | |
| Educational attainment | | | |
| Middle school and high school | 23.55 | 65 | |
| Vocational school and junior college | 47.46 | 131 | |
| University and graduate school | 28.99 | 80 | |
| | Mean (SD) | Min | Max |
| Age | 33.628 (3.44) | 24 | 38 |
| N = 274 | | | |

For the independent variable, the question posed was "Is the childcare balancing policy applicable for your workplace?" with an ordinal response of 1 = very applicable, 2 = somewhat applicable, 3 = not very applicable, and 4 = not applicable. In this analysis, the response has been coded reversely as 1 = not applicable, 2 = not very applicable, 3 = somewhat applicable, and 4 = very applicable.

Sex was coded as a binary variable—1 = male, 2 = female. For the variable "job," respondents were posed the question "Do you have a job (including student part-time job)" with a binary response choice of 1 = yes, 2 = no. The job type was a dummy variable, with the value of 1 = regular employee (typical employee, including regular employees and managers) and 0 = non-regular employee (atypical employee, including part-time job, temporary employee, contract employee, free worker, and family employee). It was renamed "regular." The question of "age" was posed as "What is your birthday?" and was coded as a continuous variable from their birth date to age, that is, 24–38 years. For "educational attainment," the question posed was "Which school did you attend last? (including the school that you are attending)." This was left as a categorical variable. However, it was changed to a dummy variable where the reference = middle and high school was named middle and high school; 1 = vocational school and junior college vocational college; 1 = university and graduate school university.

The following question was posed with six values because of the restriction item, "How satisfied are you with the relationship with your child?". When the value equals six, the respondents do not have a child. Thus, the dummy variable "child" was created, where 0 = have no children; 1 = have a child or children. For the variable "marriage," the question posed was "Are you married?" and was coded as a categorical variable with 1 = married, 2 = unmarried (single), 3 = bereavement, and 4 = divorce. To measure a respondent's income, a variable "income" was determined, including a family's income in the past year.

For the restriction item, respondents were restricted to married women who were employed. The variables "child," "marriage," and "sex" were restricted with the following values—child = 6, marriage = 1, and sex = 2.

### 3.3. Estimation Method: Binary Logistic Regression

The binary logistic regression model is an analysis in which the dependent variable is a dummy variable (Midi et al. 2010). Logistic regression measures the relationship between the categorical dependent variable and one or more categorical or continuous independent variables by estimating the probabilities using a logistic function, with a cumulative distribution (irfy[4]).

The odds in the logistic regression mean the ratio of the probability of one outcome to another, or it could be extended to explain the odds of success with one group as opposed to another, which is the odds ratio (Powers and Xie 2000).

The binary logistic regression equation appears as follows:

$$\log(Y) = \log(\frac{p}{1-p}) = \alpha + \beta_1 X_1 + \beta_2 X_2 + \ldots + \beta_n X_n \tag{1}$$

where $p$ means the probabilities of an event occurrence, $\alpha$ is the intercept of the model, $\beta$ is the coefficient of $X$, and $X$ is an independent variable (Sperandei 2014).

## 4. Results

### Analysis 1—Hypothesis 1 and 2

Table 2 shows the first binary logistic regression to examine H1 and H2 in Models 1 and 2, respectively.

**Table 2.** Binary logistic regression results predicting the possibility of women's childbearing desire using data JLPS-Y 2010.

| | Model 1 | | Model 2 | | Model 3 | |
|---|---|---|---|---|---|---|
| **Wantbaby** | **Exp(β)** | **Std. Error** | **Exp(β)** | **Std. Error** | **Exp(β)** | **Std. Error** |
| workplace | 0.608 ** | 0.096 | 0.697 * | 0.116 | 0.593 * | 0.123 |
| age | 0.804 *** | 0.04 | 0.842 ** | 0.044 | 0.804 *** | 0.04 |
| vocational collge | 2.162 * | 0.769 | 2.132 * | 0.774 | 2.165 * | 0.77 |
| university | 3.217 ** | 1.355 | 2.313 † | 1.012 | 3.206 ** | 1.352 |
| ref:middle and high school | | | | | | |
| income | 1.02 | 0.12 | 1.006 | 0.121 | 1.018 | 0.122 |
| regular employee | 2.090 * | 0.687 | 2.309 * | 0.778 | 1.751 | 1.72 |
| Ref: non-regular employee | | | | | | |
| child (ref: no child) | | | 0.181 ** | 0.089 | | |
| regular employee * workplace | | | | | 1.061 | 0.333 |
| intercept | 1640.259 | 2928.058 | 1353.643 | 2549.075 | 1813.553 | 3381.962 |
| R-square | 0.1889 | | 0.231 | | 0.189 | |
| Likelihood Ratio Test | 67.91 | | 83.01 | | 67.95 | |
| AIC | 305.7 | | 292.6 | | 307.66 | |
| BIC | 330.99 | | 321.51 | | 336.57 | |
| N = 274 | | | | | | |

*** $p < 0.001$; ** $p < 0.01$; * $p < 0.05$. † $p < 0.1$

Model 1 shows the variable "age," which, previous studies have argued, has a negative association with women's fertility intention (Roberts et al. 2011; Tsuya 1999). In other words, the odds ratio of 0.804 ($p < 0.001$) showed that with an increase in women's age, their fertility intention declined. Women with high educational attainment had a higher fertility intention than those with low educational attainment (Model 1 of Table 2). These results are contrary to most previous findings that women with high educational attainment have lower fertility intention or better-educated women favor less traditional roles (Axinn and Barber 2001; Basu 2002; Rindfuss et al. 1980). The fertility intention of women graduating from vocational college and university was higher than those graduating from middle and high school (2.162, $p < 0.05$; 3.217, $p < 0.01$, respectively). Basu (2002) showed that educated women better understood gender equality, which may influence their fertility intention. However, Heiland et al. (2005) showed that there is a positive association between desired family size and women's educational attainment. If educated women marry men who can offer an equal home, their fertility intention may not decrease.

However, this study's findings may differ from previous studies due to limited observations. Previous studies showed the relationship between income and fertility intention. Contrarily, in this study, income is not significant. Furthermore, previous studies found that female regular employees' fertility intention was higher than non-regular employees. In Model 1, the odds ratio was 2.090 ($p < 0.05$). This concurred with previous findings on the relationship between employment status and fertility intention (Murakami 2014).

Regarding the independent variable "workplace," the odds ratio was 0.608 ($p < 0.01$), which means that the association between workplace with childcare balancing policy and fertility intention was negative. In other words, the more applicable the childcare balancing policy in the workplace, the lower the fertility intention of women. However, since the condition of women without children was not limited, the objections included women who already had a child or children. In Model 2 of Table 2, women who already had a child had less fertility intention than those without children (0.181, $p < 0.01$), which concurs with previous studies that the more children they have, the less their fertility intention would be (Fukuda 2011).

Table 3 shows that women with a child or children are more likely to work in a company with a childcare balancing policy than those without children (1.092, $p < 0.001$). In Table 4, women working in a workplace with a childcare balancing policy have a higher possibility of having a child or children than those at a workplace without a policy (2.030, $p < 0.001$). However, women employed at workplaces with childcare balancing policies may already have a child or children; therefore, their fertility intention may be low.

**Table 3.** Ordinal logistic regression predicting the possibility of female working in the workplace with childcare balancing policy.

| Workplace | Exp(β) | Std. Error |
|---|---|---|
| age | −0.012 | 0.036 |
| child | 1.092 *** | 0.293 |
| ref: no child | | |
| regular employee | −0.925 *** | 0.263 |
| ref: non-regular employee | | |
| fertility desire | −0.544 * | 0.273 |
| income | −0.05 | 0.097 |
| ref: middle and high school | | |
| vocational collge | −0.267 | 0.294 |
| university | −0.570 [†] | 0.345 |
| R-square | 0.075 | |
| Likelihood Ratio Test | 53.96 | |
| N = 274 | | |

*** $p < 0.001$; * $p < 0.05$; [†] $p < 0.1$.

**Table 4.** Binary logistic regression predicting the possibility of people having child using data set JLPS-Y 2010.

| Child | Exp(β) | Std. Error |
|---|---|---|
| workplace | 2.030 *** | 0.339 |
| vocational college | 0.79 | 0.352 |
| university | 0.223 ** | 0.105 |
| ref: middle and high school | | |
| age | 1.279 *** | 0.061 |
| income | 0.887 | 0.116 |
| regular employee | 1.555 | 0.555 |
| ref:non-regular employee | | |
| intercept | 0.001 | 0.002 |
| R-square | 0.22 | |
| Likelihood Ratio Test | 70.26 | |
| N = 274 | | |

*** $p < 0.001$; ** $p < 0.01$.

In Model 3 of Table 2, to prove H2, the interaction term "workplace*regular employee" (dummy variable) was added. The variable "workplace" remained significant, although its value decreased to 0.593 ($p < 0.05$) from 0.608 ($p < 0.01$) in Model 1. Although the interaction term was added, the negative relationship between the workplace with childcare balancing policy and women's fertility intention remained. This is contrary to H1. The variable "age" showed significance although its value slightly decreased compared to Model 1, which is a negative association between age and fertility intention (0.804, $p < 0.001$). Therefore, the older the women, the lower the fertility intention, which concurs with previous research findings (Roberts et al. 2011; Tsuya 1999).

Educational attainment remained positively associated with women's fertility intention. The fertility intention of women with better education levels was higher than those graduating with lower education levels (2.165, $p < 0.05$; 3.206, $p < 0.01$, respectively in Model 3 of Table 2). Although most previous studies proved that better-educated women had less preference for traditional roles as they pursue career development (Rindfuss et al. 1996), this study's analysis showed a positive association between women's fertility intention and educational achievement. Income remained statistically insignificant.

After adding the interaction term of workplace×regular employee, the regular employee was not statistically significant in Model 3. Furthermore, the interaction term was not significant. Therefore, H2 could not be proven. There are several reasons for the non-significance of the interaction term. Non-regular employees were not qualified to benefit from childcare balancing policies in companies compared to regular employees. Furthermore, the insignificant result may be due to the limited number of observations.

## 5. Discussion and Conclusions

### 5.1. Discussion

The declining birth rate in Japan has been a serious problem. There are various reasons for explicating why the birth rate keeps decreasing, such as late marriage (Date and Shimizutani 2005) and socio-economic changes. Under this circumstance, several policies were proposed to balance individuals' work and family (Cabinet Office 2007).

This study examined the relationship between childcare balancing policy and women's fertility intention. Two hypotheses were used to predict the possibility of women's fertility intention, using the JLPS-Y data from 2010. H1 (the association between childcare balancing policy and women's fertility intention) and H2 (whether the childcare balancing policy influences regular and non-regular employees' fertility intention) were tested. The observations were restricted to married women who were employed.

For H1, the results showed that women's fertility intention was negatively associated with the childcare balancing policy in Japanese corporations, which was contrary to its

purpose (Cabinet Office 2007). A possible reason for this may be that family-friendly policies are unlikely to be strong instruments for promoting fertility (Dey 2006). These policies provide a method to balance both because childbearing has a negative association with labor participation (Bernhardt 1993). They do play an important role in women's careers, such as increasing women's employment rates (Suruga and Zhang 2003) and women in high authority positions (Yamamoto 2014; Saito 2017). This may lead to women focusing more on their work than on having a child. The study shows that the fertility intention of women who have a child or children is lower than those without children. Since the observations were not limited to women without children, there is a possibility that women who already have a child/children and work for companies with a childcare balancing policy have a low fertility intention.

Except for educational attainment, all the other results concur with previous studies. The negative relationship between the childcare balancing policy and women's fertility intention should be considered by policymakers. The reason for the negative association needs to be explored in future research. Furthermore, since the negative relationship in this study may include women who already have a child or children and work in workplaces with a childcare balancing policy, policymakers should consider the policy for married women without children. Nonetheless, for H2, the interaction term did not show significance. There are two possible explanations for the non-significance. First, non-regular employees could not benefit from the childcare balancing policy. Second, the study used a limited sample size. Given that the childcare balancing policy was proposed while most research focused on the relationship between women's employment and the policy (Abe 2007; Tomita 1994; Takeishi 2006; Saito 2017; Kawaguchi and Kasai 2013; Suruga and Zhang 2003), examining the relationship between the childcare balancing policy and women's fertility intention is an important endeavor. The findings demonstrate that the relationship between childcare balancing policy and women's fertility intention is negative, which is crucial for policymakers and the government to consider.

*5.2. Limitations*

Although this study has contributed to understanding how childcare balancing policies interact with women's fertility intentions, there are several limitations.

First, selection bias exists in this research, such as the fact that women who have higher fertility intentions tend to work in companies with childcare balancing policies. Future studies should concentrate on experimental methods to prove the causal effects between the childcare balancing policy and women's fertility intention.

Second, in dealing with the independent variable "workplace," the question posed was "Does your company reduce the work time and adjust the work hours because of childcare or study or other things?" Since the childcare balancing policy includes reduced work time and adjusting the work hours because of childcare, and there is no direct question about the childcare balancing policy, in this analysis, these independent variables were regarded as the childcare balancing policy.

Third, due to the limited sample size, only the controlled variables, important for the analysis, were added. Other variables, such as "number of siblings" and "father's involvement in childcare," were excluded.

**Funding:** This work was granted by Pioneering Research Support Project and WISE Program for AI Electronics.

**Institutional Review Board Statement:** Not applicable.

**Informed Consent Statement:** Not applicable.

**Data Availability Statement:** Data is available in a publicly accessible repository. The data presented in this study for this secondary analysis, Japanese Life Course Panel Survey of the Middle-aged (JLPS-M)and Japanese Life Course Panel Surveys (JLPS) project, Institute of Social Science, The University of Tokyo, was provided by the Social Science Japan Data Archive, Center for Social Research and Data Archives, Institute of Social Science, The University of Tokyo.

**Conflicts of Interest:** The author declares no conflict of interest.

## Notes

1    https://www.mhlw.go.jp/stf/newpage_21481.html (accessed on 13 March 2024)

2    https://www.soumu.go.jp/main_sosiki/singi/toukei/meetings/kihon_56/siryou_1j.pdf (accessed on 13 March 2024)

3    Parental leave: This is a legal leave for workers raising children (https://ja.wikipedia.org/wiki/育児休業) (accessed on 13 March 2024)

4    https://analyticsbuddhu.wordpress.com/2016/07/02/introduction-about-logistic-regression-model/ (accessed on 13 March 2024). https://medium.com/@ODSC/logistic-regression-with-python-ede39f8573c7 (accessed on 13 March 2024)

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
