# Peer review of "Childcare Balancing Policy in Japanese Corporations and Women’s Fertility Intention"

_socsci, doi:10.3390/socsci13030175_

Round 1

Reviewer 1 Report

Comments and Suggestions for Authors

-An exact definition of the childcare balancing policy studied here would be helpful to make clear upfront in the manuscript

-Suggest additional and newer citations on the trends related to late marriage

- pg. 2 (second paragraph) -Under women's employment (pg. 2- 1.3) the sources are outdated and could benefit from more and newer sources referenced. 

-How have parental leave policies changed over time? The information shared on page 4 (section 2.1.1) is very dated and I'm not sure how reflective of today's environment it is. 

-Under age (pg. 5  section 2.2.2) needs further development, detail and more relevant literature. 

-Which The Japanese Life Course Panel Survey of the Youth (JLPS-Y) data were used for this study? This isn't clear from what is stated on page 6. 

-Table 3.1 is not helpful as currently presented 

-The limitations section needs reworking for greater clarity and usability

Comments on the Quality of English Language

English is fine, but another proofing wouldn't hurt. 

Author Response

Dear Reviewer

Thanks for your valuable comments and suggestions! I learnt a lot from that.

  1. I took your advice and explained the childcare balancing policy in the front (in the beginning of introduction) , which is highlighted.
  2. I added new reference to the late marriage, which is highlighted.
  3. Under women’s employment, I added the newer source, which is highlighted.
  4. I added how the parental leave changed in 2017, like the period of paid parental leave.
  5. I added the another reference to related to age, which is highlighted. However, I am not sure that more details of age. Could you please explain that more?
  6. The Japanese Life Course Panel Survey of Youth I chose is 2010. I corrected that in the manuscript. I apologize for the misunderstanding.
  7. I revised the format of Table 3.1 as suggested.
  8. I revised the limitation as suggested, which is highlighted.

I also revised that “childcare balancing policy” as independent variable in the limitation.

The question posed was “Does your company reduce the work time and adjust the work hours because of childcare or housework or study or other things?” Since the childcare balancing policy includes reduced work time and adjusting the work hours because of childcare, and there is no direct question about the childcare balancing policy, in this analysis, these independent variables were regarded as the childcare balancing policy.

I got my manuscript English editing.

Reviewer 2 Report

Comments and Suggestions for Authors

The paper focuses on the relationship between childcare balancing policy and women's fertility intentions in Japan. The topic is important and relevant for present-day policy makers in Japan and elsewhere. The paper needs to be improved in the following areas:

1. Include more recent literature covering reasons for declining birth rate in developed countries (see e.g. Comolli et al 2020 on the role of economic and welfare uncertainty)

2. Include more detailed description of Japanese family policy context (see e.g. Kohara & Maity 2021) and especially the non-statutory "childcare balancing policy" applied in corporations.

3. Clarify the information on the data and methods:

3.1. it is somewhat unclear which year (2009 or 2010) was used in the analysis.

3.2. describe what "regular" and "non-regular" employment means in the Japanese context.

4. The results section includes quite a lot of interpretation and conclusions, and the conclusion section includes quite a lot of description of results. For better clarity, these could be separated so that in the results section, empirical results are described and in the discussion section, the results are interprated in relation to previous research, and policy recommendations are given.

Comments on the Quality of English Language

The English Language needs polishing, please use a professional language editor.

Author Response

Dear Reviewer

Thanks for your valuable suggestions and comments and the references you recommended. I learned a lot from that.

  1. I added the more recent literature related to declining birth rate in developed countries, which is highlighted. However, I did not find the reference you mentioned. Could you please tell me the name of the paper?
  2. I added Kohara and Maity (2021)’s reference about family-friendly policies in the text. However, I could not find the literature about how the childcare balancing policy (non-statutory) applied in the company. I added the rate of companied having the childcare balancing policy.
  3. The year of data I chose is 2010. I apologize for the misunderstanding.
  4. The “regular” and “irregular” is like as typical and atypical employee. I added more information in the manuscript, which is highlighted.
  5. I separated the results and discussion part.

I also revised that “childcare balancing policy” as independent variable in the limitation.

The question posed was “Does your company reduce the work time and adjust the work hours because of childcare or housework or study or other things?” Since the childcare balancing policy includes reduced work time and adjusting the work hours because of childcare, and there is no direct question about the childcare balancing policy, in this analysis, these independent variables were regarded as the childcare balancing policy.

I got my manuscript English editing.

Round 2

Reviewer 2 Report

Comments and Suggestions for Authors

The paper has improved as the comments have been taken into account, and the English language is much better after language editing.